# Feasibility of Multimodal Deep Learning for Automated Staging of Familial Exudative Vitreoretinopathy Using Color Fundus Photographs and Fluorescein Angiography

**DOI:** 10.3390/diagnostics15212752

**Published:** 2025-10-30

**Authors:** Mingzhen Yuan, Tianyu Wang, Zirong Liu, Jinghua Liu, Jing Ma, Guangda Deng, Liang Li, Songfeng Li, Yan Hu, Hai Lu

**Affiliations:** 1Beijing Tongren Eye Center, Beijing Tongren Hospital, Beijing Ophthalmology and Visual Sciences Key Laboratory, Capital Medical University, Beijing 100730, China; 2The Research Institute of Trustworthy Autonomous Systems and the Department of Computer Science and Engineering, Southern University of Science and Technology, Shenzhen 518055, China

**Keywords:** familial exudative vitreoretinopathy, deep learning, multimodal, fundus photography, fluorescein angiography, staging, decision-support

## Abstract

**Introduction:** To evaluate the feasibility of multimodal deep learning (DL) for automated staging of familial exudative vitreoretinopathy (FEVR) using color fundus photographs (CFP) and fluorescein angiography (FFA). **Methods:** We assembled a multimodal dataset across FEVR stages 0–5 and post-laser cases and benchmarked CNNs (Convolutional Neural Networks), Transformers, and multimodal fusion under center-region and multi-image settings. Class imbalance was mitigated via weighted sampling and focal/class-balanced losses. We report accuracy, recall, precision, macro-F1, Cohen’s κ, and class-wise ROC/AUC with 95% Cis. **Results:** AI system showed balanced performance versus specialists (0.65 vs. Dr. A: 0.48/Dr. B: 0.48) in CFP assessment, maintaining high specificity (0.91–0.92). Among architectures: (1) Transformers outperformed CNNs in single-modal analysis; (2) ResNet showed moderate performance (AUC 0.70–0.85) but limited capability for intermediate grades (AUC < 0.70); (3) CRD-Net achieved peak performance (AUC up to 0.94, severe cases AUC > 0.90). While FFA improved Dr. B’s accuracy to 0.56, it remained below AI levels. Stage-specific accuracy ranged from 0.72 to 0.88 across the FEVR spectrum. **Conclusions:** Leveraging a novel multimodal database and high-performance AI models, systematic comparisons demonstrated the superiority of Transformer architectures over CNNs in single-modal analysis, while CRD-Net’s multimodal fusion approach achieved optimal performance across all severity grades. Multimodal DL shows feasibility as a decision-support tool for automated FEVR staging within confirmed cohorts.

## 1. Introduction

Familial exudative vitreoretinopathy (FEVR) was first described by Criswick and Schepens in 1969. It is a hereditary ocular disorder characterized primarily by incomplete peripheral retinal vascular development, neovascularization, and exudation [1]. With the widespread implementation of neonatal fundus triage/decision-support, the detection rate of FEVR in newborns can reach 0.63–1.19% [2], and FEVR accounts for 13–16% of blinding eye diseases in children [3,4]. Due to the high clinical heterogeneity and diverse disease staging of FEVR [5,6], significant subjectivity and considerable variability exist among physicians in its diagnosis and classification [7]. As a result, FEVR patients are prone to misdiagnosis and underdiagnosis, leading to missed optimal treatment windows. This may progress to severe ocular complications such as retinal detachment, ultimately resulting in permanent blindness [8,9]. Therefore, there is an urgent need to develop a triage/decision-support tool capable of rapidly identifying fundus images that require further attention and rigorous analysis by ophthalmologists, thereby improving diagnostic accuracy and efficiency.

Deep learning (DL) is a mature and continuously evolving technology, particularly in the field of computer-aided diagnosis (CAD) for human diseases [10,11]. DL has made significant contributions to the automated triage/decision-support and diagnosis of various retinal diseases, with some technologies already implemented in clinical practice [12,13,14,15]. Among pediatric retinal diseases, DL has been most extensively applied to retinopathy of prematurity (ROP) [16,17,18,19]. This is largely due to the widespread adoption of routine postnatal fundus examinations in preterm infants, facilitating the collection of ROP fundus color photographs. Additionally, since infants with ROP are often medically unstable, fluorescein fundus angiography (FFA) is typically unavailable during early triage/decision-support. Consequently, ROP diagnosis and staging based on color fundus photography (CFP) alone have gained clinical acceptance [20]. Despite being one of the most common pediatric retinal diseases and sharing clinical manifestations/staging similarities with ROP, FEVR critically lacks dedicated datasets to support algorithm development and experimental research [21,22]. The current limitations in FEVR research arise from two major challenges: First, there remains a lack of comprehensive understanding and sufficient clinical emphasis on FEVR in practice. Second, the diagnostic evaluation and staging of FEVR necessitates the combined use of CFP and FFA [23]. However, the clinical acquisition of both imaging modalities simultaneously presents significant practical difficulties, which substantially hinders the establishment of comprehensive FEVR datasets and consequently impedes the development of intelligent diagnostic assistance systems.

In this study, we established the first comprehensive multimodal FEVR dataset, encompassing both CFP and FFA images across the disease spectrum (Stages 0–5 and post-laser treatment). To lay a robust foundation for future research, we implemented various imaging-based automated classification models for FEVR, designed for precise detection and staging. The models’ diagnostic accuracy was rigorously validated through comparative analysis with assessments from experienced retinal specialists. This pioneering research demonstrates the potential for automated FEVR detection and staging with favorable accuracy and specificity, offering clinical decision-support capabilities and advancing the development of personalized treatment strategies for FEVR.

## 2. Methods

### 2.1. Ethics Approval

This study was approved by the Ethics Committee of Beijing Tongren Hospital, Capital Medical University (Approval No.: TREC2022-XJS05) and strictly adhered to the principles of the Declaration of Helsinki throughout the research process. Written informed consent was obtained from all participating patients or their legal guardians prior to enrollment. Furthermore, all personally identifiable information was removed from the imaging data prior to analysis to ensure complete anonymization and confidentiality of participant information.

### 2.2. Datasets

This study retrospectively analyzed 2268 CFP and 4740 FFA images from 244 eyes of 154 FEVR patients diagnosed at Beijing Tongren Hospital, Capital Medical University, between 1 January 2013, and 31 December 2024. The diagnosis was confirmed by experienced retinal specialists based on comprehensive clinical evaluations including ocular characteristics, birth history, and family history. All imaging was performed using the Retcam III wide-angle digital imaging system (Clarity Medical Systems, Pleasanton, CA, USA) for both color fundus photography and sodium fluorescein angiography. FFA procedures and subsequent treatments were conducted under continuous monitoring by qualified anesthesiologists to ensure patient safety.

In this study, the diagnosis of FEVR was based not solely on imaging features but primarily on genetic testing as the gold standard. All included patients were genetically confirmed to have FEVR through sequencing. Targeted genetic testing for FEVR-associated genes was performed, including FZD4, LRP5, TSPAN12, NDP, ZNF408, KIF11, and CTNNB1, consistent with current literature [24,25,26,27]. Under this diagnostic framework, we classified cases as Stage 0 when genetic confirmation was achieved despite the absence of abnormal clinical signs—such as avascular zones, leakage, or vascular traction—in the relatively healthy eye, as assessed by both CFP and FFA [28]. The clinical staging of FEVR was conducted in accordance with the system established by Pendergast et al. (refer to Table 1 and Figure 1) [29]. The study enrolled patients who were either treatment-naïve or had only undergone retinal laser photocoagulation (Figure 1). Exclusion criteria were as follows: (1) coexistence of other retinal diseases, including Norrie disease, retinopathy of prematurity, persistent fetal vasculature, or other heritable retinal disorders; (2) history of ocular trauma or intraocular surgery; (3) presence of severe systemic comorbidities; (4) poor-quality imaging data featuring decentration, exposure anomalies (over- or underexposure), or significant artifacts; and (5) cases where staging was not feasible due to insufficient or discrepant CFP and FFA results.

The image collection protocol mandated a minimum of 5 standardized CFP images per FEVR case: macular-centered posterior pole (encompassing both macula and optic disc) along with superior, inferior, nasal, and temporal peripheral retinal images, with FFA images following identical anatomical coverage requirements. All images were systematically evaluated, with CFP documenting retinal lesions and FFA assessing: (1) optic disc morphology/margins, (2) retinal vascular course/morphology, (3) macular vascular arcade integrity/dimensions, and (4) peripheral avascular zones, neovascularization, retinal detachment, and abnormal fluorescence patterns. To ensure diagnostic reliability, two pediatric retinal specialists (YMZ and MJ) independently performed FEVR staging, with any discrepancies resolved through adjudication by a third senior specialist (LH).

### 2.3. Image Labeling, Preprocessing and Dataset Division

During the initial triage/decision-support of CFP and FFA images for analysis, several types of low-quality images were excluded to ensure data reliability. These included: (1) Excessively Blurred Images: Pictures with significant defocus or motion blur that hindered clear visualization of retinal structures, such as blood vessels, exudates, or avascular zones. (2) Poor Illumination: Underexposed or overexposed images where key features were obscured due to uneven lighting or flash artifacts. (3) Incomplete Field of View: Images that did not fully capture the required retinal area (e.g., missing peripheral regions critical for FEVR staging). (4) Artifacts and Obstructions: Photos affected by eyelash shadows, dust on the lens, or severe media opacities that interfered with retinal assessment. (5) Misalignment or Cropping Issues: Improperly centered images where the optic disc or macula was cut off, preventing standardized analysis. After this triage/decision-support step, we retained 2268 CFP and 4740 FFA images for analysis. According to our records, only a small portion (approximately 5–10%) of the originally collected images were excluded due to quality issues. By training and testing our models on this high-quality dataset, we aimed to simulate a realistic clinical setting where only diagnostically acceptable images are used.

In this study, we established seven labels (0, 1, 2, 3, 4, 5, 6) with the following definitions: label 0 represents Stage 0 FEVR; label 1 represents Stage 1 FEVR; label 2 represents Stage 2 FEVR; label 3 represents Stage 3 FEVR; label 4 represents Stage 4 FEVR; label 5 represents Stage 5 FEVR; label 6 indicates prior retinal laser photocoagulation treatment. The specific diagnostic criteria for FEVR staging are detailed in Table 2.

The dataset division strategy was based on the number of patient eyes to ensure strict separation between training and test sets, thus avoiding potential data leakage issues. The specific division method was as follows:

We implemented an 8:2 ratio division based on the number of patient eyes (80% for training, 20% for testing). During this division process, several factors were taken into consideration:Ensuring that images of both eyes from the same patient would not simultaneously appear in both training and test sets.Maintaining balanced distribution of disease severity across both datasets.Preserving equilibrium of different age groups and gender distributions between the two datasets.

In total, our dataset comprised images from 244 eyes, with 195 eyes (80%) allocated to the training set and 49 eyes (20%) assigned to the test set. Each eye included paired CFP and FFA images, allowing the model to learn the correlations between these two imaging modalities.

### 2.4. Development of the Algorithm

This exploratory study was designed to systematically evaluate existing methodologies in the context of FEVR assessment using a novel dataset. Rather than developing new algorithms, we focused on establishing baseline performance metrics of well-established convolutional neural networks (CNNs) and transformer-based architectures when applied to our specific clinical context. The primary objective was to provide a comprehensive benchmark that could serve as a foundation for future algorithmic developments in this domain.

We constructed a multi-faceted experimental framework to examine the efficacy of different deep learning architectures across varying imaging modalities and region-of-interest approaches. The investigation centered on two key dimensions: (1) comparison between center-focused single images versus multi-image approaches, and (2) evaluation of single-modality CFP analysis versus integrated CFP-FFA assessment methods. This design allowed us to systematically address the question of whether the additional complexity of multiple images or dual-modality inputs yields clinically significant improvements in diagnostic accuracy when processed through competitive deep learning models.

The research paradigm adhered to established clinical validation practices, emphasizing reproducibility and clinical relevance while leveraging contemporary deep learning architectures. This approach was deemed appropriate given that this represents the first systematic investigation using this particular dataset and clinical context, thus prioritizing robust baseline establishment across multiple algorithmic families.

Our experimental protocol was structured to isolate and quantify the effects of different imaging approaches and modality combinations while exploring various deep learning architectures. The experiment comprised four distinct configurations:Center-region CFP analysis: Implementation of deep learning algorithms on center-focused CFP images only, reflecting the most streamlined clinical workflow.Multi-image CFP analysis: Application of identical algorithms to multiple CFP images per eye, incorporating peripheral retinal information.Center-region CFP-FFA combined analysis: Integration of corresponding center-focused CFP and FFA images through multi-modal fusion techniques.Multi-image CFP-FFA combined analysis: Comprehensive assessment utilizing multiple images from both CFP and FFA modalities.

For each configuration, we employed a diverse set of contemporary deep learning architectures, as shown in Figure 2, including CNN-based, Transformer-based and multimodal-based algorithms. For experiments utilizing only CFP images, we implemented several single-modality algorithms. For CNN-based algorithms, as shown in Figure 2a, *ResNet-18*, a relatively lightweight residual network with 18 layers, was selected to evaluate whether less complex models could achieve satisfactory performance with reduced computational requirements. *ResNet-50*, a deeper residual network offering enhanced feature extraction capabilities through its 50-layer architecture, provided a more robust feature representation. We also explored transformer-based approaches, as shown in Figure 2a, including *ViT-B* (Vision Transformer-Base), a transformer architecture that processes image patches as sequences and leverages self-attention mechanisms to capture global dependencies. *ConvNeXt-V2*, a modern convolutional network incorporating architectural improvements inspired by transformers while maintaining the inductive biases of CNNs, was implemented to leverage recent advances in CNN design. Additionally, *FlexiViT*, a variable-resolution vision transformer that adapts its patch size based on input complexity, was utilized to potentially better handle varying levels of detail in retinal images.

For integrated CFP-FFA analysis, we implemented several multi-modal fusion algorithms, as shown in Figure 2a. *MM-CNN* served as a foundational approach, processing each modality through separate convolutional pathways before combining features at multiple levels. *CRD-Net* provided a more sophisticated fusion mechanism, incorporating cross-modal attention mechanisms to identify complementary information between the CFP and FFA modalities. *MSAN* represented our most advanced fusion approach, integrating attention mechanisms at different feature resolutions to simultaneously capture fine-grained lesion details and broader contextual patterns across both imaging modalities.

The implementation parameters for each method were determined through preliminary experiments on a separate validation subset and subsequently standardized across all configurations to ensure fair comparison. All models were initialized with weights pre-trained on ImageNet, followed by comprehensive fine-tuning on our training dataset. Network training employed standard optimization techniques including the Adam optimizer with an initial learning rate of 10^−4^, cosine annealing scheduling, and early stopping based on validation performance.

Data augmentation techniques including random rotations, flips, and intensity adjustments were uniformly applied across all models to enhance generalization. There is no explicit exposure correction performed in the experiments. We employed a cross-entropy loss function with class weighting to address potential class imbalance in the training data.

Performance evaluation employed a standardized set of metrics including specificity, accuracy, recall, f1_score, kappa and area under the receiver operating characteristic curve (AUC-ROC). Additionally, we conducted statistical comparison between ROC curves to quantify the significance of observed performance differences between configurations and architectures. To mitigate potential bias from the train–test split, all evaluation metrics were computed using the test dataset that was strictly segregated from the training process.

The experimental design incorporated appropriate controls for potential confounding factors such as image quality variations, patient demographic characteristics, and disease severity distribution. Statistical analysis of results included confidence interval calculation and hypothesis testing to establish the statistical significance of observed performance differences between different architectures and data configurations.

All experiments were conducted on an NVIDIA RTX 2080Ti GPU. The models were trained with a batch size of 64 for 100 epochs. We employed an initial learning rate of 0.001 with either Adam optimizer or SGD (with momentum 0.9), selected based on empirical performance across different model architectures. Cross-entropy loss was used as the objective function for all models.

Regarding input dimensions, most models utilized a standard input size of 224 × 224 pixels, while FlexiViT required a slightly larger input size of 240 × 240 pixels due to its architectural specifications. For data augmentation, we implemented random rotation (up to 180 degrees) to enhance model generalization. All images were normalized using the standard ImageNet normalization parameters (mean = [0.485, 0.456, 0.406], std = [0.229, 0.224, 0.225]) to maintain consistency with established preprocessing protocols.

### 2.5. Statistical Analysis

All analyses were performed using SPSS 22.0 (IBM Corp). Continuous variables with normal distribution were expressed as mean ± standard deviation (SD), while non-normally distributed data were presented as median (Q1, Q3). Correlation analyses employed the Spearman method, with *p* < 0.05 considered statistically significant.

## 3. Results

### 3.1. Patient Information

This study included 154 pediatric FEVR patients, comprising 100 males (100/154, 65.94%) and 54 females (54/154, 35.06%). The mean age of the entire cohort was 3.45 ± 2.20 years (range: 2 months to 15 years). The mean ages stratified by FEVR stage were as follows: Stage 1: 3.99 ± 2.45 years (range: 6 months to 11 years); Stage 2: 4.21 ± 2.57 years (range: 9 months to 15 years); Stage 3: 3.29 ± 1.92 years (range: 2 months to 7 years); Stage 4: 2.66 ± 1.51 years (range: 5 months to 6 years); Stage 5: 2.93 ± 2.23 years (range: 5 months to 6 years). It should be emphasized that some enrolled patients presented with discordant FEVR stages between eyes. For the purpose of demographic and statistical analyses, the most advanced stage observed in either eye was adopted as the representative classification.

### 3.2. Overall Performance Comparison

The experimental results from Table 3 demonstrate varying performance across different deep learning architectures for FEVR grading, with notable differences between center region and multi-image approaches. Among all tested models, CRD-Net consistently achieved the highest performance metrics across both test configurations, suggesting its superior capability in capturing disease-relevant features for FEVR classification.

### 3.3. Center Region Analysis

For the center region test set, CRD-Net achieved the highest accuracy (0.72), precision (0.81), and F1 score (0.72), with a substantial Kappa coefficient of 0.66, indicating good agreement beyond chance. FlexiViT demonstrated the second-best performance with balanced metrics (accuracy: 0.64, F1 score: 0.57, Kappa: 0.57), suggesting its effectiveness in handling single-region analysis. Notably, all models maintained high specificity (ranging from 0.75 to 0.83), indicating robust performance in correctly identifying negative cases.

The transformer-based architectures (ViT-B and FlexiViT) showed competitive performance compared to traditional CNN architectures, with FlexiViT outperforming ViT-B across all metrics. Among the ResNet variants, ResNet50 slightly outperformed ResNet18, though both achieved moderate performance levels. MSAN demonstrated the lowest performance with an accuracy of 0.50 and Kappa of 0.40, suggesting limitations in its architecture for this specific task.

### 3.4. Multi-Image Analysis

The multi-image approach generally resulted in decreased performance across most models, indicating the increased complexity of integrating information from multiple retinal images. However, CRD-Net maintained relatively stable performance (accuracy: 0.63, F1 score: 0.60, Kappa: 0.55), demonstrating its robustness to varying input configurations. Interestingly, MM-CNN showed improved relative performance in the multi-image setting compared to other models, suggesting its architecture may be better suited for multi-view feature integration.

### 3.5. Clinical Significance

The recall values, crucial for disease triage/decision-support applications, varied considerably across models. In the center region analysis, CRD-Net achieved the highest recall (0.69), followed by MM-CNN and FlexiViT (0.59). This high sensitivity is particularly important for FEVR detection, where missing positive cases could have serious clinical consequences. The generally lower recall values in the multi-image setting (ranging from 0.42 to 0.62) suggest that the added complexity may introduce challenges in maintaining sensitivity.

### 3.6. Multi-Class ROC Performance

The Receiver Operating Characteristic (ROC) curves, as shown in Figure 3, demonstrate the discriminative capability of each model across different FEVR severity grades. Each model was evaluated using a one-vs-rest approach, generating individual ROC curves for each disease class. The Area Under the Curve (AUC) values reveal significant variations in class-specific performance across all tested architectures.

CRD-Net exhibited the most robust performance with consistently high AUC values across all classes, achieving peak AUC values of 0.94 for certain severity grades. This superior discriminative ability suggests that CRD-Net’s architecture effectively captures the subtle morphological variations characteristic of different FEVR stages. The model demonstrated particularly strong performance in distinguishing severe cases (AUC > 0.90), which is crucial for timely clinical intervention.

### 3.7. Class-Specific Performance Patterns

Analysis of the ROC curves reveals interesting class-specific patterns. Most models showed higher AUC values for extreme severity grades (both mild and severe), while intermediate grades proved more challenging to classify. This pattern aligns with clinical observations that borderline cases often present ambiguous features. FlexiViT and MM-CNN demonstrated more balanced performance across classes, with AUC values ranging from 0.72 to 0.91, suggesting their architectures are less susceptible to class imbalance effects.

Notably, transformer-based models (ViT-B and FlexiViT) showed competitive performance with CNN-based architectures, particularly for classes with subtle feature differences. FlexiViT’s adaptive patch selection mechanism appeared to contribute to improved discrimination, as evidenced by its consistently higher AUC values compared to standard ViT-B across most severity grades.

### 3.8. Model-Specific Observations

ResNet architectures (ResNet18 and ResNet50) displayed moderate performance with AUC values predominantly in the 0.70–0.85 range. While ResNet50 showed marginal improvements over ResNet18, both models struggled with certain intermediate severity grades (AUC < 0.70), suggesting limitations in their feature extraction capabilities for subtle retinal changes.

ConvNeXt-v2, despite its modern architecture, showed variable performance across classes, with some categories achieving high AUC values (>0.85) while others remained moderate. This variability suggests that while the model captures certain disease patterns effectively, it may require further optimization for comprehensive FEVR grading.

MSAN demonstrated the lowest overall performance with several classes showing AUC values below 0.60, indicating limited discriminative capability for this specific application. This underperformance across multiple severity grades suggests fundamental limitations in its architecture for retinal image analysis.

### 3.9. Attention Mechanism Analysis

To further understand model decision-making, we conducted attention visualization experiments on representative cases from each severity grade, as shown in Figure 4 and Figure 5.

The attention maps revealed distinct patterns across different architectures: FlexiViT’s flexible attention patterns adapted to image-specific features, allocating higher attention weights to regions with vascular abnormalities while maintaining global context. This adaptive behavior explains its robust performance across diverse presentations of FEVR. Traditional CNN models (ResNet variants and MM-CNN) showed more diffuse attention patterns, often focusing on central retinal regions despite the peripheral nature of FEVR pathology. This mismatch between attention focus and disease localization partially explains their moderate performance.

CRD-Net’s attention mechanisms consistently focused on clinically relevant regions, including the peripheral retina where FEVR changes typically manifest, and vascular branching points critical for disease assessment. This alignment with clinical expertise validates the model’s learned representations.

### 3.10. Clinical Implications

The high AUC values achieved by top-performing models (particularly CRD-Net with AUC > 0.90 for several classes) demonstrate the potential of automated FEVR grading. However, further improvements in other evaluation metrics, such as accuracy, recall, and F1-score, are necessary to achieve clinical applicability. The ability to reliably distinguish severe cases (high sensitivity at clinically relevant specificity thresholds) supports the potential use of these systems for triage/decision-support applications. The attention visualization results provide interpretability crucial for clinical adoption, allowing clinicians to understand and verify model decisions. The correlation between model attention and known disease patterns enhances trust in automated predictions.

### 3.11. Performance at Clinical Operating Points

Analysis of the ROC curves at clinically relevant operating points (e.g., 95% specificity) reveals that CRD-Net maintains sensitivity above 0.80 for most severity grades, meeting requirements for triage/decision-support applications. This performance level suggests the model could effectively identify cases requiring specialist referral while minimizing false positives. These comprehensive results demonstrate that deep learning approaches, particularly architectures designed for medical imaging like CRD-Net, can achieve clinically meaningful performance for FEVR grading. The combination of high discriminative ability and interpretable attention patterns supports the potential integration of these systems into clinical workflows for improved FEVR diagnosis and management.

### 3.12. Clinical Evaluation

To assess whether the deep learning model has attained diagnostic capability comparable to human experts, we conducted a rigorous evaluation using External Test Set B with two pediatric retina specialists of varying experience levels: Dr. A (0–5 years’ experience in pediatric retinal diseases) and Dr. B (over 5 years’ experience). Following a double-blind protocol where neither clinician had access to patient histories or each other’s assessments, both physicians independently evaluated the test set comprising 50 eyes (50 color fundus photographs paired with 50 corresponding fluorescein angiography images), all featuring 130-degree macular-centered retinal images.

To contextualize these automated results, we examined the diagnostic performance of two experienced clinicians (Dr. A and Dr. B) on the center region dataset. The data revealed interesting comparative insights between human experts and our computational models:

When using conventional visual assessment (CFP), both clinicians achieved an accuracy of 0.48, which is lower than our best-performing automated systems. However, they demonstrated high specificity scores (0.91), indicating excellent ability to correctly identify negative cases. Dr. A showed slightly higher recall (0.52) and precision (0.59) compared to Dr. B’s recall (0.51) and precision (0.54).

Remarkably, when clinicians utilized the feature attention-assisted approach (CFP-FFA), Dr. B’s performance improved substantially, with accuracy increasing to 0.56, recall to 0.58, and precision to 0.60. The specificity also marginally improved to 0.92, with F1 score reaching 0.54 and kappa coefficient 0.47. In contrast, Dr. A showed minimal improvement under the feature attention-assisted approach, maintaining similar metrics as with conventional assessment. See Table 4 for details.

## 4. Discussion

FEVR is an inherited retinal vascular dysplasia disease characterized by avascular areas, abnormal vascular proliferation and exudative changes in the peripheral retina. It is one of the most common inherited retinopathy in children. Its clinical staging (stage 0–5) directly affects treatment decisions, for example: Early stage (stage 1–2): close observation or preventive laser treatment is required. Intermediate stage (3): management is individualized—laser/anti-VEGF for active neovascularization, with consideration of scleral buckle and/or pars plana vitrectomy when tractional or extramacular retinal detachment is present. Late stage (stage 4–5): Urgent surgical intervention is required to prevent retinal detachment [9]. However, due to the diversity of FEVR fundus lesions and the uneven level of doctors in treating pediatric fundus diseases, the staging standards are highly subjective, which makes it easy for doctors to miss the early tiny avascular areas by naked eye assessment, with a missed diagnosis rate of up to 30% [30]. Therefore, this study developed a deep learning (DL) system for automated staging of FEVR using multimodal retinal images (CFP and FFA). Our results demonstrate that the DL model achieves balanced performance comparable to senior ophthalmologists, while significantly outperforming junior clinicians and traditional rule-based methods.

Early studies from 2018 to 2020 mainly used FFA images as a single data modality. Although they achieved an accuracy rate of 82%, they ignored practical problems such as the invasiveness of FFA examinations and the poor cooperation of pediatric patients. After 2021, the research entered the multimodal fusion stage, and the combined CFP and FFA became the mainstream method, with the accuracy rate increased by 12% [23]. Some cutting-edge research began to try to integrate OCTA technology. In terms of model architecture, from the development of the basic CNN model to the introduction of the attention mechanism, and then to the attempt of the Transformer architecture, the model performance has been continuously improved but the data demand has also increased accordingly [13]. Our study demonstrates that integrating FFA with CFP significantly enhances the diagnostic accuracy of FEVR staging compared to CFP alone. This finding corroborates the established pathophysiology of FEVR, as FFA provides critical dynamic visualization of retinal perfusion status, clearly delineating avascular zones, vascular leakage patterns, and early neovascularization—features that are frequently undetectable on conventional fundus imaging. Although the accuracy values may appear modest, our model’s overall performance in internal validation was comparable to that of senior retinal specialists and significantly outperformed junior clinicians. This highlights the highly subjective and complex nature of FEVR staging itself and suggests that even an imperfect AI system can serve as a valuable standardized auxiliary tool.

The above comprehensive analysis indicates that the feature attention-enhanced contrastive learning strategy demonstrates significant advantages in medical image analysis, with the CRD-Net model paired with this strategy showing particularly outstanding performance. From a clinical utility perspective, this methodology enables more precise capture of key pathological features in medical images, thereby improving diagnostic accuracy. Transformer-based architectures (such as ViT-b and FlexiViT) also demonstrate promising potential, suggesting that these novel architectures may bring further breakthroughs in the field of medical imaging analysis. These findings provide important guidance for developing more precise and reliable computer-aided diagnostic systems, with the potential to assist clinicians in improving diagnostic efficiency and accuracy in clinical practice, ultimately benefiting patients.

The performance gap between center region and multi-image approaches highlights the challenge of effectively integrating information from multiple retinal views. Future work should focus on developing specialized architectures that can better leverage multi-view information while maintaining high sensitivity for disease detection. Additionally, the moderate Kappa values (maximum 0.66) suggest room for improvement in achieving expert-level agreement, which could be addressed through enhanced feature extraction methods or ensemble approaches. These results provide valuable insights for clinical implementation, suggesting that CRD-Net with center region analysis offers the most reliable performance for FEVR grading in current clinical workflows, while highlighting the potential for further optimization in multi-image integration strategies.

Moreover, the study focused on a single 130° central macular image for FEVR staging because this is the most practical and widely used imaging modality in the triage/decision-support setting. Our results showed that the DL model outperformed clinicians in diagnostic accuracy, especially in distinguishing early FEVR (stages 1–3), as human observers may overlook these subtle vascular abnormalities. Moreover, deep learning models demonstrate superior sensitivity to subtle features in FEVR diagnosis. For instance, characteristic findings such as peripheral avascularity or mild exudation often present with low contrast in fundus images. While human graders must rely on subjective interpretation of these faint signs, the DL model utilizes hierarchical feature extraction to identify minute pathological patterns—including microvascular anomalies and early neovascularization—that may be imperceptible to the human eye. These findings demonstrate that our deep learning system can achieve reliable staging performance using only a single macular-centered image. This demonstrates that machine learning can effectively extract clinically significant biomarkers from posterior pole features, such as reduced vascular arcade angle and macular dragging signs, which serve as reliable proxies for disease severity assessment. By detecting these subtle indicators, the deep learning model enables earlier referral for confirmatory wide-field imaging or therapeutic intervention when necessary. Importantly, the AI model’s comparable and, in certain aspects, superior performance underlines its promise as a valuable diagnostic aid in routine triage/decision-support and long-term follow-up settings, helping standardize evaluations and reduce diagnostic variability especially when specialized imaging is not readily accessible.

Several factors may explain why our multi-image (multi-view) models did not outperform the single-image models. First, there is a complex tradeoff between redundancy and complementarity of information. The central CFP and FFA images contain overlapping anatomical information, so a model might rely mostly on the more salient CFP features (e.g., clearly visible vessels) and underutilize the modalities’ unique signals (e.g., peripheral non-perfusion or early leakage seen only on FFA). If the network cannot effectively attend to these FFA-specific indicators of disease (such as avascular zones or subtle dye leakage), then simply adding the extra image views will add little useful information. In support of this, Temkar et al. note that RetCam-assisted FFA is “extremely useful to document peripheral retinal vascular pathologies” that are often missed on CFP [31]. In other words, the potential benefit of FFA data depends on the model’s ability to draw out the complementary features rather than just repeating what is already in the CFP.

Second, our current fusion architecture may not fully capture the complex, nonlinear relationships between modalities. CFP and FFA images of the same region are not spatially aligned and represent lesions very differently (for example, a peripheral avascular area appears bright on FFA but only as a faint gray patch on CFP). A simple feature concatenation or shallow fusion layer may fail to align these semantic correspondences across modalities. More advanced fusion strategies—such as attention-based cross-modal transformers—have been proposed to address this issue. For instance, Liu et al. introduce a cross-modal attention module that adaptively focuses on lesion-related features across different imaging modalities, improving the integration of complementary information from CFP and FFA [32]. Our experience suggests that without such specialized fusion mechanisms, the added complexity of multi-image input may not translate into higher classification accuracy.

Finally, it is useful to compare our findings with related studies. Some prior retinal imaging work has shown benefits from multi-view inputs. For example, Boualleg et al. demonstrate that multi-view deep feature fusion significantly improves classification accuracy in diabetic retinopathy [33]. However, disease-specific factors likely play a role: diabetic retinopathy often produces more prominent central lesions and large datasets are available, whereas FEVR features are more subtle and peripheral. In addition, our multimodal FEVR dataset, while sizable for this rare disease, is relatively limited, especially for advanced stages. These differences in data and pathology may explain why we did not observe a similar performance gain. In summary, the combination of overlapping image information, limitations of our current fusion method, and the particular characteristics of FEVR likely contributed to the lower performance of the naive multi-image models. Future work using more sophisticated fusion architectures and larger, multi-center datasets may better exploit the complementary information in multiple fundus views.

The FEVR staging model developed in this study showed promising accuracy but has several limitations. It was designed specifically for staging disease severity rather than initial diagnosis, and was trained exclusively on confirmed FEVR cases. The dataset also included relatively few Stage 5 images, potentially introducing staging bias, and all data originated from a single center with homogeneous population characteristics and equipment, which may limit generalizability to other clinical settings.

Future efforts will focus on developing a more comprehensive system capable of both differential diagnosis and staging. This will require a larger-scale FEVR image dataset incorporating data from diverse populations and clinical environments, supported by advanced data augmentation techniques and multi-institutional collaboration to improve the detection of rare and advanced-stage cases. Further testing and optimization of sensitivity metrics will be necessary to minimize false-negative rates. Additionally, we plan to integrate multimodal clinical metadata—such as birth history, medical and family history, and genetic test results—to enhance the AI-based diagnosis. Subsequent studies will utilize datasets from multiple clinical centers and larger patient cohorts to validate and optimize this system for practical potential decision-support utility. Moreover, advanced model optimization techniques, such as ensemble learning, transfer learning, and domain-specific fine-tuning, will be explored to improve performance on complex and imbalanced data distributions. Integrating auxiliary tasks, such as retinal segmentation or vascular abnormality detection, within a multi-task learning framework may also provide valuable contextual information to improve overall classification accuracy.

## 5. Conclusions

We curated a comprehensive multimodal FEVR dataset and evaluated deep-learning models for automated staging. The multimodal system achieved balanced performance on this confirmed-FEVR cohort and, on several metrics, performed comparably to pediatric retina specialists, supporting its feasibility as a decision-support tool rather than a population-level screening method. Given class imbalance and the lack of non-FEVR controls, further work should include multi-center external validation, differential-diagnosis cohorts, operating-point optimization prioritizing sensitivity, and calibration/decision-curve analyses. With these steps, multimodal AI could contribute to more standardized staging and longitudinal follow-up in specialized settings.

## Figures and Tables

**Figure 1 diagnostics-15-02752-f001:**
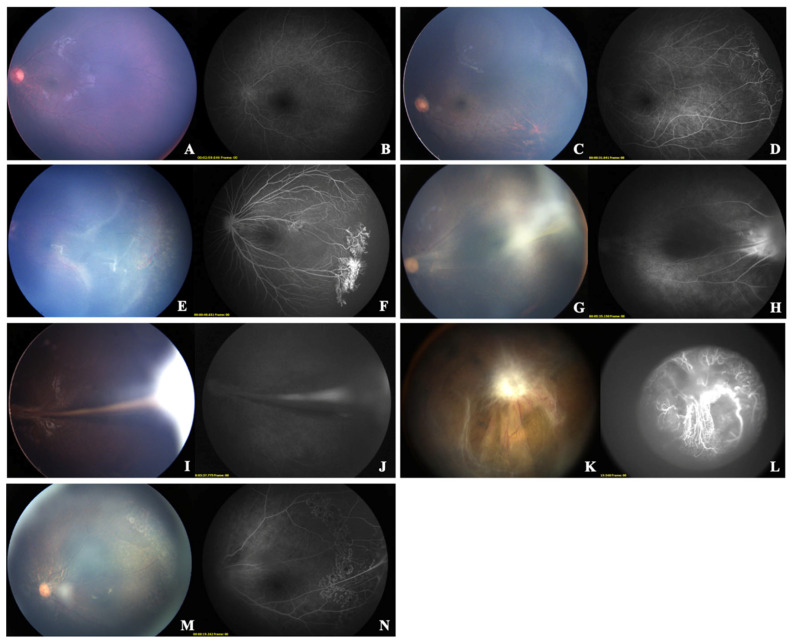
Representative CFP and FFA Images Across FEVR Stages: (**A**,**B**): Stage 0 FEVR (CFP and FFA). (**C**,**D**): Stage 1 FEVR (CFP and FFA). (**E**,**F**): Stage 2 FEVR (CFP and FFA). (**G**,**H**): Stage 3 FEVR (CFP and FFA). (**I**,**J**): Stage 4 FEVR (CFP and FFA). (**K**,**L**): Stage 5 FEVR (CFP and FFA). (**M**,**N**): Post-laser photocoagulation in FEVR (CFP and FFA).

**Figure 2 diagnostics-15-02752-f002:**
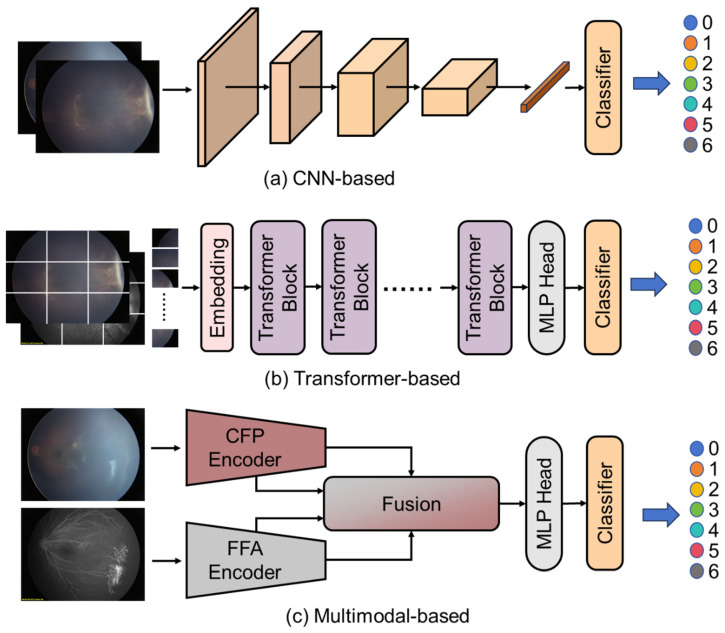
Deep learning-based algorithms.

**Figure 3 diagnostics-15-02752-f003:**
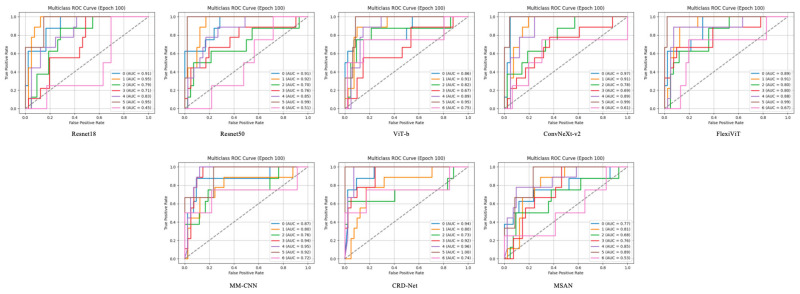
AUC Example Results.

**Figure 4 diagnostics-15-02752-f004:**
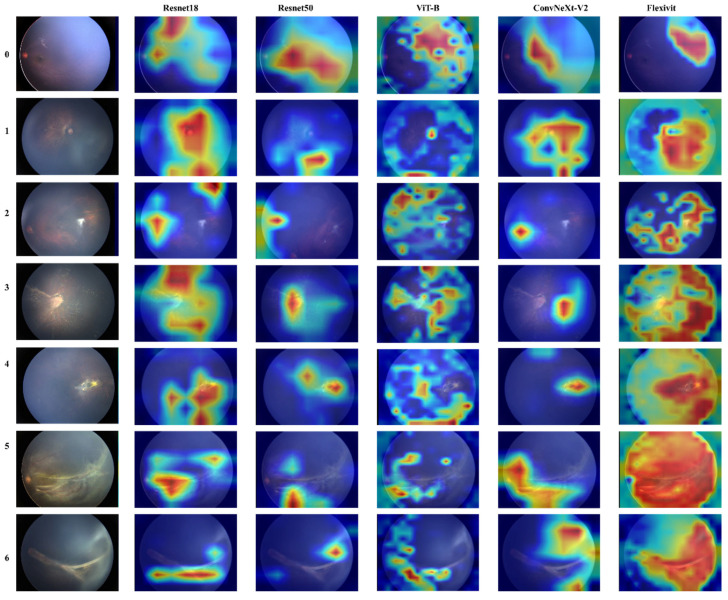
Attention map of different grades by different single-modal algorithms.

**Figure 5 diagnostics-15-02752-f005:**
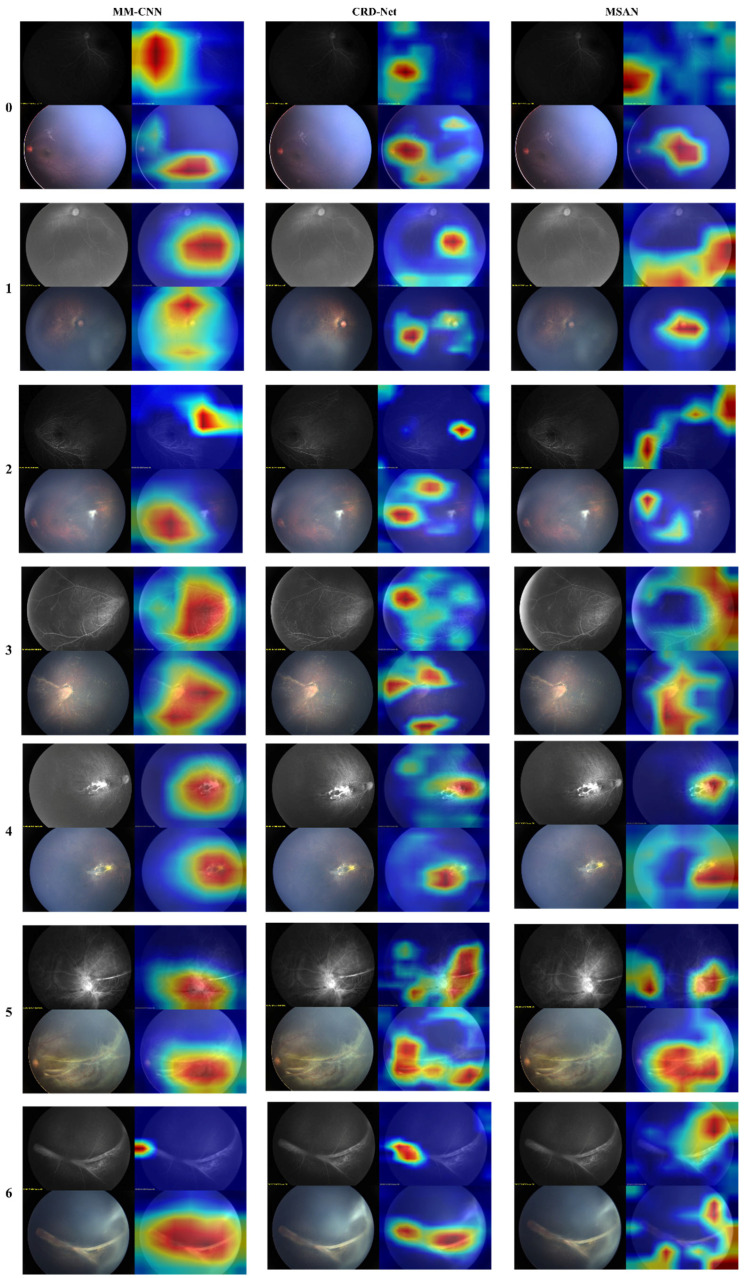
Attention map of different grades by different multi-modal algorithms.

**Table 1 diagnostics-15-02752-t001:** Staging Criteria for FEVR.

Stage	Description
Stage 0	A completely normal fundus or the presence of retinal vascular abnormalities (visible loops and increased branching) without peripheral avascular zones
Stage 1	The presence of a peripheral avascular zone without neovascularization
Stage 2	The development of retinal neovascularization
Stage 3	The presence of retinal detachment not involving the macula
Stage 4	Retinal detachment involving the macula
Stage 5	Total retinal detachment

**Table 2 diagnostics-15-02752-t002:** Summary of Research Labels.

Label	0	1	2	3	4	5	6	Total
Pairs	40	45	37	45	45	15	17	244
CFP	360	457	323	387	449	106	186	2268
FFA	446	994	935	990	821	196	358	4740

**Table 3 diagnostics-15-02752-t003:** The Model’s Performance on the Different Test Sets.

			Accuracy	Recall	Precision	Specificity	F1_Score	Kappa
Center region	**CFP**	resnet18	0.52	0.49	0.50	**0.83**	0.47	0.42
		resnet50	0.56	0.49	**0.58**	**0.83**	0.49	0.47
		ViT-b	0.56	0.51	0.55	**0.83**	0.51	0.47
		ConvNeXt-V2	0.54	0.48	0.51	0.75	0.45	0.44
		FlexiViT	**0.64**	**0.59**	0.57	**0.83**	**0.57**	**0.57**
	**CFP-FFA**	MM-CNN	0.60	0.59	0.68	0.57	0.61	0.52
		CRD-Net	**0.72**	**0.69**	**0.81**	**0.75**	**0.72**	**0.66**
		MSAN	0.50	0.46	0.56	0.50	0.47	0.40
Multi-image	**CFP**	resnet18	0.48	0.44	0.47	0.72	0.43	0.38
		resnet50	0.49	0.45	0.52	0.72	0.44	0.38
		ViT-b	**0.54**	**0.50**	**0.64**	0.63	**0.51**	**0.45**
		ConvNeXt-V2	0.44	0.42	0.54	**0.86**	0.41	0.33
		FlexiViT	0.53	0.48	0.61	0.76	0.50	0.43
	**CFP-FFA**	MM-CNN	0.56	0.58	**0.63**	**0.63**	0.57	0.47
		CRD-Net	**0.63**	**0.62**	**0.63**	0.60	**0.60**	**0.55**
		MSAN	0.45	0.43	0.45	0.47	0.42	0.33

**Table 4 diagnostics-15-02752-t004:** Diagnostic Performance of Pediatric Retinal Specialists.

			Accuracy	Recall	Precision	Specificity	f1_Score	Kappa
Center region	**CFP**	Dr. A	0.48	0.52	0.59	0.91	0.53	0.38
		Dr. B	0.48	0.51	0.54	0.91	0.48	0.38
	**CFP-FFA**	Dr. A	0.48	0.49	0.57	0.91	0.50	0.38
		Dr. B	0.56	0.58	0.60	0.92	0.54	0.47

## Data Availability

The dataset used and analyzed during the current study are available from the corresponding author on reasonable request.

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
