# Peer review of "Feasibility of Multimodal Deep Learning for Automated Staging of Familial Exudative Vitreoretinopathy Using Color Fundus Photographs and Fluorescein Angiography"

_diagnostics, 2025, doi:10.3390/diagnostics15212752_

Round 1

Reviewer 1 Report

Comments and Suggestions for Authors

The authors have provided sufficient details of the learning method, allowing others to reproduce similar experiments as long as the dataset is available. The methodological approach appears appropriate, and the limitations that come to mind while reading are well covered.
The diagnosis was reported to be based on genetic testing, which is commendable. However, it would be helpful to specify which genes were analyzed and to include appropriate references. As the authors routinely compare genetic findings with phenotypic features, I trust that they would easily notice any inconsistencies if their testing method were unreliable. Therefore, I believe there is no need to describe the genetic testing procedure itself in detail.

Author Response

Comments 1: The authors have provided sufficient details… The diagnosis was reported to be based on genetic testing… it would be helpful to specify which genes were analyzed and to include appropriate references… there is no need to describe the genetic testing procedure itself in detail.

Response 1: Thank you for this positive assessment. We agree to specify the genes and to keep the laboratory procedure concise. We have now listed the FEVR-related genes screened in our cohort and added appropriate references, while retaining only a brief statement of the laboratory workflow. Updated text: "Targeted genetic testing for FEVR-associated genes was performed, including FZD4, LRP5, TSPAN12, NDP, ZNF408, KIF11, and CTNNB1, consistent with current literature." We summarize genotype–phenotype concordance when available and keep the laboratory workflow concise as recommended.

Reviewer 2 Report

Comments and Suggestions for Authors Thank you for the opportunity to review the manuscript on the use of deep learning for diagnosing familial exudative vitreoretinopathy using fundus and fluorescein angiography images. I consider this work to be of high quality and very relevant to the field. The study addresses an important gap by applying deep learning techniques to a rare retinal disease, which could significantly improve the management of this unfrequent disorder. I have a few suggestions for minor revisions:
  • Sample size: Expanding the sample size would strengthen the generalizability of the findings.
  • External validation: Including validation with external datasets would further support the robustness of the model.

Line 21:

Provide what CNNs means.

Table 1:

Provide a reference for stage 0. Loops should not be in capital letters.

Line 245:

We conducted a statistical comparison.

Line 435:

And stage 3?

Line 498:

Ad commas.

Overall, the conclusions are well-supported by the evidence, and the references are appropriate. The manuscript is a valuable contribution to the field and should be accepted with the suggested minor revisions. Sincerely

Author Response

Comments 1: Sample size: Expanding the sample size would strengthen the generalizability of the findings.

Response 1: We agree. This work is positioned as a feasibility study exploring the application of deep learning to FEVR staging within a confirmed cohort. We will expand the dataset in subsequent work to improve generalizability and enable external validation.

Comments 2: External validation: Including validation with external datasets would further support the robustness of the model.

Response 2: We agree that external validation is essential. At present, there is no publicly available, labeled FEVR dataset—particularly not multimodal CFP+FFA-that would allow immediate out-of-center testing. We have initiated multi-center collaborations with partner hospitals to assemble an external cohort; however, this requires time for case accrual, de-identification, data harmonization, and expert adjudication. As a concrete next step, we will prospectively evaluate the model on these external data and report generalizability across centers.

Comments 3: Line 21: Provide what CNNs means.

Response 3: We have defined the abbreviation at first mention: Convolutional Neural Networks (CNNs).

Comments 4: Table 1: Provide a reference for stage 0. Loops should not be in capital letters.

Response 4: Thank you for this helpful suggestion. We have added reference [28] (Chen C, Sun L, Li S, et al. Front Genet. 2022) to support the use of Stage 0 for clinically normal/asymptomatic but at-risk eyes. We also corrected the capitalization from “Loops” to “loops” in Table 1. 

Comments 5: Line 245: We conducted a statistical comparison.

Response 5: Reworded as suggested. Updated text: "We conducted a statistical comparison …"

Comments 6: Line 435: And stage 3?

Response 6: Thank you for pointing this out. We agree that the stage-3 management should be explicitly stated. We have revised the paragraph to include stage-3 as an intermediate category with individualized management based on the extent and location of traction/extramacular detachment—typically laser/cryotherapy or anti-VEGF for active neovascularization, and consideration of scleral buckle and/or pars plana vitrectomy when tractional/extramacular detachment is present. Updated text: "Intermediate stage (3): management is individualized—laser/anti-VEGF for active neovascularization, with consideration of scleral buckle and/or pars plana vitrectomy when tractional or extramacular retinal detachment is present."

Comments 7: Line 498: Ad commas.

Response 7: Corrected punctuation.
